# RDDM: Practicing RAW Domain Diffusion Model for Real-world Image Restoration

**Yan Chen**[1*], **Yi Wen**[1*], **Wei Li**[1†], **Junchao Liu**[1], **Yong Guo**[2], **Jie Hu**[1], **Xinghao Chen**[1]
[1] Huawei Noah's Ark Lab
[2] Max Planck Institute for Informatics
{chenyan176, wenyi14, wei.lee}@huawei.com

(a) Real-World Image Restoration Results Starting from Sensor RAW

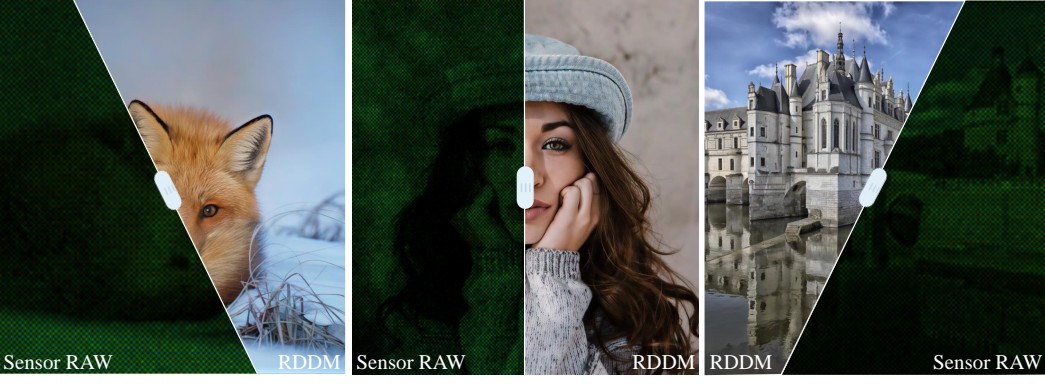

Sensor RAW — RDDM | Sensor RAW — RDDM | RDDM — Sensor RAW

(b) Comparison with Two-Stage ISP+IR Models

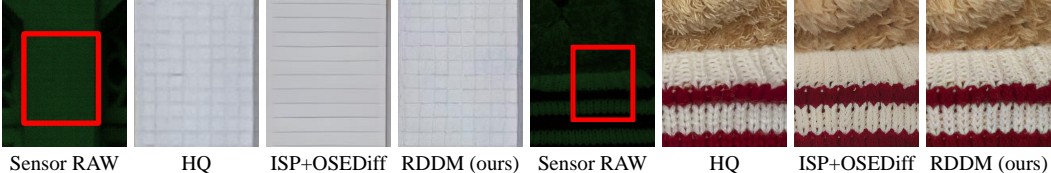

Sensor RAW · HQ · ISP+OSEDiff · RDDM (ours) · Sensor RAW · HQ · ISP+OSEDiff · RDDM (ours)

Figure 1: RDDM, restoring directly from the sensor RAW data, demonstrates remarkable results shown in (a), capitalizing on the unprocessed and detail-rich signal. Compared with the two-stage baseline in (b), RDDM delivers markedly higher fidelity and perceptual quality.

## ABSTRACT

We present the RAW domain diffusion model (RDDM), an end-to-end diffusion model that restores photo-realistic images directly from the sensor RAW data. While recent sRGB-domain diffusion methods achieve impressive results, they are caught in a dilemma between high fidelity and realistic generation. As these models process lossy sRGB inputs and neglect the accessibility of the sensor RAW images in many scenarios, e.g., in image and video capturing in edge devices, resulting in sub-optimal performance. RDDM obviates this limitation by directly restoring images in the RAW domain, replacing the conventional two-stage image signal processing (ISP)→Image Restoration (IR) pipeline. However, a simple adaptation of pre-trained diffusion models to the RAW domain confronts the out-of-distribution (OOD) issues. To this end, we propose: (1) a RAW-domain VAE (RVAE), encoding sensor RAW and decoding it into an enhanced linear domain image, (2) a configurable multi-bayer (CMB) LoRA module, adapting diverse RAW Bayer patterns such as RGGB, BGGR, etc. To compensate for the deficiency in the dataset, we develop a scalable data synthesis pipeline synthesizing RAW LQ-HQ pairs from existing sRGB datasets for large-scale training. Extensive experiments demonstrate RDDM's superiority over state-of-the-art sRGB diffusion methods, yielding higher fidelity results with fewer artifacts. Codes are publicly available at github.com/YanCHEN-fr/RDDM.

---

*Equal Contribution.
†Project Lead.

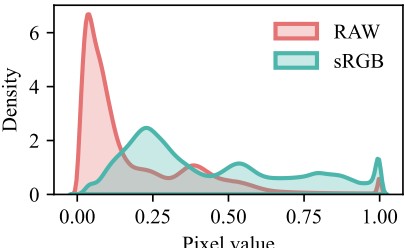
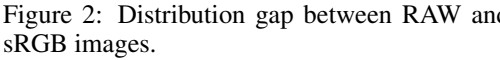



(a) DIV2K-Val          (b) RealSR

Figure 2: Distribution gap between RAW and sRGB images.

Figure 3: The performance comparison among SD-based methods on test datasets DIV2K-Val and RealSR, respectively.

# 1 INTRODUCTION

Real-world Image Restoration (Real-IR) aims to restore high-quality (HQ) images from low-quality (LQ) images containing complex degradations, e.g. noise, image compression and blur (Fan et al., 2020; Jinjin et al., 2020; Zhang et al., 2022; 2019; 2018b; 2017). Existing GAN-based (Ledig et al., 2017; Wang et al., 2018) methods employ a generator and a discriminator for adversarial training. However, GAN-based methods suffer from pattern collapse, incurring unsatisfactory results (Liang et al., 2022a; Chen et al., 2022; Liang et al., 2022b; Xie et al., 2023). Benefiting from the powerful generative priors granted by text-to-image (T2I) models, SUPIR (Yu et al., 2024) and its counterparts (Zhang et al., 2023; Lin et al., 2024; Wang et al., 2024; Wu et al., 2024b; Yu et al., 2024) integrate pre-trained diffusion models into Real-IR and have attained remarkable performance, which prevails Real-IR models process and enhance images in the sRGB domain. Despite the success of T2I methods in the sRGB domain, they still face the challenge on the dilemma between image fidelity and realistic generation.

In edge-device imaging, the ISP processes the sensor RAW capture into an sRGB image, compressing dynamic range from 12-14 bit into an 8 bit and discarding details through operations like demosaicing, AWB, and clipping, leading to sub-optimal restoration results (Nguyen & Brown, 2016; Xing et al., 2021). This drives us to integrate T2I models, which possess extensive priors, into the RAW domain to leverage the unprocessed native signal. However, this integration faces challenges: the significant differences in luminance, mosaic patterns, and noise distribution between sRGB and RAW images render sRGB-pretrained models ineffective for RAW processing, as shown in Fig. 1, and a comprehensive RAW-domain Real-IR dataset for rigorous benchmarking is currently lacking. Fig. 2 exhibits the RAW-sRGB distribution gap.

In this paper we propose a novel RAW Domain Diffusion Model (RDDM) that starts from sensor RAW to fully exploit the image information lost during the ISP process (Sundararajan, 2017), thereby unlocking enhanced potential for visual restoration effects. To bridge the RAW-sRGB distribution gap, we firstly devise and train a RVAE by a divide-and-conquer strategy that accepts a sensor RAW as the input and outputs a enhanced linear domain image. Additionally, we design a CMB LoRA module for Bayer pattern adaptability. To further mitigate OOD issues, we design a PTP module that enables joint RAW and sRGB space optimization, thereby improving model fidelity. Finally, to addressing data scarcity, we propose a RAW image synthesis method, allowing us to synthesize abundant RAW–linear image pairs from publicly accessible sRGB datasets.

In summary, our main contributions are as follows:

- We propose RDDM, the first practical application of the raw domain diffusion model, establishing a novel paradigm for RAW image restoration.

- We propose RVAE capable of encoding mosaicked, noised RAW images and subsequently decoding the latent representations into linear HQ images, resolving the OOD issues. Additionally, we design a RAW domain Real-IR data synthesis method and construct a RAW Real-IR benchmark.

- Extensive experiments verify that RDDM demonstrates superior image fidelity and comparable generation capability to the state-of-the-art methods, as illustrated in Fig. 3

## 2 RELATED WORK

**Real-world Image Restoration.** Real-IR is becoming a trending field of research since the advent of ESRGAN (Ledig et al., 2017). Early studies attempted various ways to combine generative adversarial networks (GANs) (Goodfellow et al., 2014; Karras et al., 2017; 2019; Radford et al., 2015; Mirza & Osindero, 2014) with perceptual losses (Ding et al., 2020; Johnson et al., 2016; Zhang et al., 2018a) for training networks to predict images that follow the natural image distribution (Ledig et al., 2017; Wang et al., 2018; 2021; Zhang et al., 2021; Liang et al., 2022a; Chen et al., 2022; Liang et al., 2022b; Xie et al., 2023). However, since adversarial training of GANs can be unstable, their discriminators are deficient in determining the quality of the diverse natural image contents, giving rise to unnatural visual artifacts. As an alternative to GAN-based methods, diffusion-based models (Podell et al., 2023; Rombach et al., 2022) is becoming increasingly popular in Real-IR tasks (Kawar et al., 2022; Li et al., 2022; Luo et al., 2023a;b; Özdenizci & Legenstein, 2023; Saharia et al., 2022) to generate realistic images with substantial texture, leveraging pre-trained Stable Diffusion (SD) models as priors (Zhang et al., 2023; Lin et al., 2024; Wang et al., 2024; Wu et al., 2024b; Yu et al., 2024; Sun et al., 2024; Menon et al., 2020; Karras et al., 2019) whereas they employ different condition injection strategies and feature extraction. Nevertheless, all existing Real-IR methods restore images in the sRGB domain in which rich information in the RAW domain might be lost after ISP. However, directly adapting sRGB Real-IR methods to the RAW domain encounters severe domain mismatch and results in poor performance.

**Image Processing in RAW Domain.** Edge-devices capture sensor RAW data. The ISP pipeline reconstructs sRGB images from sensor RAW through sequential hand-crafted modules, including demosaicing (DM), denoising (DN), automated-white-balance (AWB), color correction matrix (CCM), gamma compression (GC), and tone mapping (TM) (Sundararajan, 2017). After the demosaicing module of ISP, a RAW domain image is transformed into a linear image, which is then converted to an sRGB image after the subsequent post tone processing. One-stage RAW domain image processing methods (Brooks et al., 2019; Cao et al., 2024; Li et al., 2024; Qian et al., 2019) typically integrate denoising and demosaicing, which can result in image oversmoothing. The processed images are then converted to sRGB images through the Post tone-mapping modules (AWB, CCM, GC, TM), where operations such as AWB and clipping compress image information and further cause detail loss in sRGB images. For two-stage methods (ISP→IR), performing image restoration on these lossy sRGB images often leads to artifacts.

## 3 METHODOLOGY

### 3.1 PROBLEM MODELING

In the sRGB domain, Real-IR model $G_\theta^{rgb}$, parameterized by $\theta$, aims to estimate HQ sRGB image $\hat{X}_H^{rgb} \in \mathbb{R}^{h \times w \times 3}$ given LQ sRGB image $X_L^{rgb} \in \mathbb{R}^{h \times w \times 3}$. In RAW domain, we train a neural network $G_\theta^{RAW}$ to transform LQ sensor RAW $X_L^{RAW} \in \mathbb{R}^{h \times w \times 1}$ to HQ linear domain image $\hat{X}_H^{lin} \in \mathbb{R}^{h \times w \times 3}$. The training task can be modeled as the following optimization problem:

$$\theta^* = argmin_\theta \mathbb{E}_{X_L^{RAW}, X_H^{lin} \sim S} \left[ \mathcal{L}(G_\theta^{RAW}(X_L^{RAW}, \epsilon), X_H^{lin}) \right] \tag{1}$$

where S is the dataset consisting of $(X_L^{RAW}, X_H^{lin})$ pairs, and $\mathcal{L}$ is the loss function, respectively.

A feed-forward ISP typically transforms RAW or linear domain images into the sRGB color space, comprising a joint denoising and demosaicing module followed by a Post Tone Processing (PTP) module. Fig. 5 (a) presents an overview of a feed-forward ISP. We train a lightweight joint denoising and demosaicing network $\mathcal{F}_{DD}(\cdot)$ to produces a linear domain image $\hat{X}_H^{lin}$ given a sensor RAW $X_L^{RAW}$, can be formulated as:

$$\hat{X}_H^{lin} = \mathcal{F}_{DD}(X_L^{RAW}) \tag{2}$$

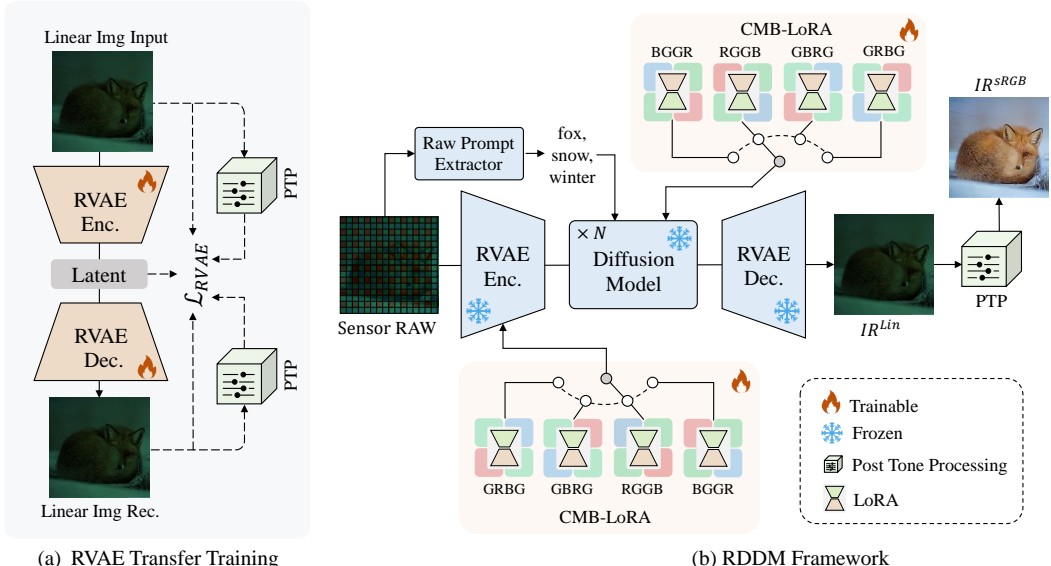

(a) RVAE Transfer Training       (b) RDDM Framework

Figure 4: (a) Illustration of RVAE training strategy. (b) With the adapted RVAE, we jointly train the CMB-LoRA of RVAE encoder and pre-trained diffusion network using RAW-linear image pairs.

where $\mathcal{F}_{DD}$ is optimized by minimizing: $\mathcal{L}_{DDNet} = ||(\mathcal{F}_{DD}(X_L^{RAW}), X_H^{lin})||_2^2$. Post Tone Processing (PTP) module $\mathcal{F}_{PTP}(\cdot)$ including AWB, CCM, GC and TM, converts linear domain images $\hat{X}_H^{lin}$ to sRGB images $\hat{X}_H^{rgb}$, which is defined as:

$$\hat{X}_H^{rgb} = \mathcal{F}_{PTP}(\hat{X}_H^{lin}), \tag{3}$$

Since the on-device ISP pipeline is typically a black box, this simplified implementation is introduced solely to validate the efficacy of the proposed Real-IR network. A detailed introduction of ISP, as well as their mathematical derivations referenced in the Appendix A.1.

## 3.2 RAW DOMAIN DIFFUSION MODEL

**Framework Overview**. Our network $G_\theta^{RAW}$ is composed of an RVAE encoder $E_\theta^{lin}$, a diffusion network $\epsilon_\theta$, and an RVAE decoder $D_\theta^{lin}$. $E_\theta^{lin}$ encodes the noisy, mosaicked sensor RAW into a latent representation. $\epsilon_\theta$ refines this latent code to recover fine detail. $D_\theta^{lin}$ then decodes the enhanced features to yield a linear, demosaiced and denoised image that is converted to sRGB by the PTP module. We incorporate trainable LoRA layers (Hu et al., 2022) into the pre-trained $E_\theta^{lin}$ and $\epsilon_\theta$. To address the issue of extracting prompts from sensor RAW images, the feed-forward ISP firstly processes the sensor RAW image to obtain an sRGB image, from which the DAPE (Zheng et al., 2024) prompt extractor extracts the textual information to activate priors. Fig. 4 presents an overview of the framework and the interplay between the various modules.

**RAW Domain VAE.** VAE plays a pivotal role in the quality of generated images. However, existing VAE in the sRGB domain are not capable of effectively encoding RAW images and decoding linear domain images. Therefore, we train a RAW domain VAE that encodes RAW images and subsequently decodes the latent representation into a linear domain image. (Rombach et al., 2022) employs a scaling factor to normalize the latent space distributions of different VAEs to the standard Gaussian distribution, which is beneficial for diffusion network optimization. To obtain the statistically accurate scaling factor, we calculate the parameter for training samples in the linear domain according to the following formula:

$$\sigma^2 = \frac{1}{bchw} \sum_{b,c,h,w} (z^{b,c,h,w} - \hat{\mu}), \hat{\mu} = \frac{1}{bchw} \sum_{b,c,h,w} z^{b,c,h,w} \tag{4}$$

where $z^{b,c,h,w}$ denotes the latent space of the training samples encoded by $E_\theta^{lin}$. $\hat{\mu}$ and $\sigma^2$ present the mean and variance of the data distribution. The rescaled latent has unit standard deviation, i.e.,

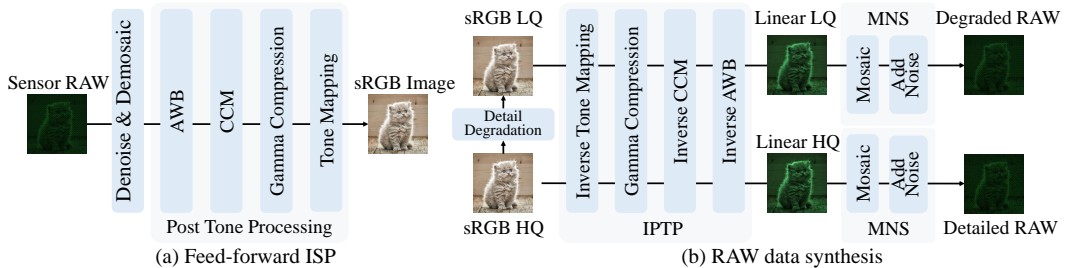

(a) Feed-forward ISP                    (b) RAW data synthesis

Figure 5: Feed-forward ISP and RAW data synthesis pipeline. IPTP transforms an sRGB image into its linear domain counterpart. MNS converts a linear domain image into a sensor RAW image.

$z \leftarrow \frac{z}{\sigma}$. The training strategy of RVAE is illustrated in Fig. 4 (a). We train the encoder and decoder on the linear domain dataset degraded from DIV2K, Flickr2K, LSDIR, DIV8K and the first 10K face images from FFHQ using our data synthesis pipeline, such that the linear domain input $X_H^{lin}$ is encoded by RVAE encoder to obtain the latent feature $z$ and the RVAE decoder decodes $z$ into the target linear domain image $\hat{X}_H^{lin} = D(E(X_H^{lin}))$. Furthermore, we devise a differentiable PTP module that simultaneously supervises training in both the sRGB and RAW domains. Similar to LDM (Rombach et al., 2022), we use $L_1$ loss, LPIPS loss, and GAN loss to train the VAE encoder and decoder to generate realistic details of a linear image:

$$\mathcal{L}_{RVAE} = L_{rec}(\hat{X}_H^{lin}, X_H^{lin}) + \lambda_G L_{GAN}(\hat{X}_H^{lin}, X_H^{lin}) \qquad (5)$$

where $L_{rec} = L_1 + L_{LPIPS}$ and $L_1$ is calculated in both the RAW and sRGB domains. $\lambda_G = \frac{\nabla[L_{rec}]}{\nabla[L_{GAN}]+10^{-4}}$ and $\nabla[\cdot]$ represents the gradient of the last layer in the decoder. In order to accommodate RAW images captured with arbitrary Bayer patterns, we propose a configurable multi-Bayer (CMB) LoRA module that augments the RVAE encoder and the pre-trained diffusion network with independent sets of LoRA, and we assign a distinct LoRA group to each Bayer pattern.

**Training Framework.** During training, the RVAE decoder is frozen and only the CMB LoRA modules are optimized. We use VSD, LPIPS loss, and MSE loss to train our model in the RAW domain and the sRGB domain:

$$\mathcal{L} = L_{VSD}(\hat{X}_H^{lin}, X_H^{lin}) + \lambda_1 L_{RAW}(\hat{X}_H^{lin}, X_H^{lin}) + \lambda_2 L_{rgb}(\mathcal{F}_{PTP}(\hat{X}_H^{lin}), \mathcal{F}_{PTP}(X_H^{lin})) \qquad (6)$$

where $\lambda_1, \lambda_2$ are weighting scalars. $L_{RAW} = L_{MSE}$. $L_{rgb} = L_{MSE} + L_{LPIPS}$. We transform both the model predictions and the ground-truth into the sRGB domain via the proposed PTP module.

### 3.3 DATA SYNTHESIS

Despite the abundance of existing datasets for Real-IR, such as LSDIR (Li et al., 2023), FFHQ (Karras et al., 2019), and DIV2K (Agustsson & Timofte, 2017), these datasets are all in the sRGB domain. To the best of our knowledge, there is currently no dataset for Real-IR in the RAW domain. Therefore, to provide a solid training foundation for Real-IR in the RAW domain, we synthesize a RAW domain Real-IR dataset by degrading publicly available sRGB Real-IR datasets, as shown in Fig. 5 (b). In particular, we first degrade sRGB HQ images $X_H^{rgb}$ to sRGB LQ images $X_L^{rgb}$ via detail degradation method, following Real-ESRGAN (Wang et al., 2021) despite excluding the degradation process of random noise, since the noise in the RAW domain is intrinsic in the sensor's physical features. We devise a inverse post tone processing module (IPTP) $\mathcal{F}_{PTP}^{-1}(\cdot)$, transforming sRGB domain image $\hat{X}_H^{rgb}$ to linear domain image $\hat{X}_H^{lin}$, and a mosaic noise synthesizer module (MNS) $\mathcal{F}_{DD}^{-1}(\cdot)$, transforming $\hat{X}_H^{lin}$ to RAW image $X_L^{RAW}$, which is defined as:

$$\hat{X}_H^{lin} = \mathcal{F}_{PTP}^{-1}(\hat{X}_H^{rgb}), X_L^{RAW} = \mathcal{F}_{DD}^{-1}(\hat{X}_H^{lin}) \qquad (7)$$

for the synthesis of the training dataset for RDDM, the sRGB LQ images are processed through $\mathcal{F}_{PTP}^{-1}(\cdot)$ and $\mathcal{F}_{DD}^{-1}(\cdot)$ to obtain degraded RAW images, while the sRGB HQ images are transformed into linear domain GT $X_H^{lin}$ through $\mathcal{F}_{PTP}^{-1}(\cdot)$. For the synthesis of the training dataset for DDNet, linear HQ images $X_H^{lin}$ are processed through $\mathcal{F}_{DD}^{-1}(\cdot)$ to produce detailed RAW images.

Table 1: Quantitative comparison with different methods on both synthetic benchmarks. The best, second best and third results of each metric are highlighted by red , orange and yellow cells respectively. ↓ presents the smaller the better, ↑ presents the bigger the better. Please note that we denote the number of sampling steps for each diffusion-based method using the format "method-steps".

| Dataset | Method | PSNR↑ | SSIM↑ | LPIPS↓ | DISTS↓ | FID↓ | NIQE↓ | MUSIQ↑ | CLIPIQA↑ |
|---|---|---|---|---|---|---|---|---|---|
| DIV2K-Val | JDnDmSR | 23.4565 | 0.6192 | 0.5347 | 0.2655 | 45.3706 | 7.0895 | 32.1252 | 0.1978 |
| | SwinIR | 22.7983 | 0.6294 | 0.5345 | 0.2780 | 44.9270 | 7.1012 | 32.9053 | 0.2520 |
| | MambaIRv2 | 23.6377 | 0.6009 | 0.5882 | 0.2749 | 42.8625 | 7.2673 | 31.6576 | 0.2010 |
| | ISP+StableSR-s200 | 23.6034 | 0.6133 | 0.4095 | 0.2092 | 35.6300 | 4.7840 | 43.8325 | 0.4284 |
| | ISP+DiffBIR-s50 | 22.4903 | 0.5284 | 0.4519 | 0.2176 | 42.0167 | 4.6040 | 52.9640 | 0.6503 |
| | ISP+PASD-s20 | 23.3860 | 0.6150 | 0.3029 | 0.1385 | 23.5801 | 3.4392 | 64.3181 | 0.6197 |
| | ISP+SeeSR-s50 | 23.2836 | 0.6059 | 0.2880 | 0.1363 | 25.4424 | 3.5605 | 65.6650 | 0.6976 |
| | ISP+SUPIR-s50 | 22.4837 | 0.5935 | 0.3265 | 0.1462 | 27.4418 | 3.5376 | 62.7078 | 0.5570 |
| | ISP+OSEDiff-s1 | 22.5277 | 0.6069 | 0.2836 | 0.1351 | 38.0461 | 3.6427 | 66.2024 | 0.6818 |
| | Ours-s1 | 23.7416 | 0.6296 | 0.2540 | 0.1197 | 23.8028 | 3.3627 | 65.4202 | 0.6737 |
| DRealSR | JDnDmSR | 27.6972 | 0.7995 | 0.3610 | 0.2210 | 31.1697 | 7.9294 | 30.4728 | 0.2373 |
| | SwinIR | 27.0657 | 0.8161 | 0.3714 | 0.2305 | 30.5639 | 7.5234 | 30.2268 | 0.2972 |
| | MambaIRv2 | 28.5563 | 0.7655 | 0.4409 | 0.2359 | 27.8182 | 8.1602 | 29.1201 | 0.2595 |
| | ISP+StableSR-s200 | 27.1173 | 0.7613 | 0.3387 | 0.1978 | 25.8442 | 4.5959 | 49.2604 | 0.5991 |
| | ISP+DiffBIR-s50 | 28.2670 | 0.7606 | 0.4142 | 0.2702 | 25.9530 | 6.3725 | 38.1396 | 0.5284 |
| | ISP+PASD-s20 | 28.3377 | 0.7845 | 0.2870 | 0.1670 | 16.1714 | 4.6875 | 53.1539 | 0.5872 |
| | ISP+SeeSR-s50 | 27.6513 | 0.7765 | 0.2972 | 0.1816 | 19.2938 | 4.2053 | 56.0800 | 0.6681 |
| | ISP+SUPIR-s50 | 26.9559 | 0.7359 | 0.3262 | 0.1799 | 26.1866 | 5.0892 | 48.5114 | 0.4839 |
| | ISP+OSEDiff-s1 | 25.1101 | 0.7315 | 0.3396 | 0.1900 | 32.4002 | 4.7336 | 57.3375 | 0.7376 |
| | Ours-s1 | 28.3495 | 0.7892 | 0.2719 | 0.1649 | 17.4825 | 4.6852 | 57.0696 | 0.7035 |
| RealSR | JDnDmSR | 25.6346 | 0.7532 | 0.3649 | 0.2119 | 66.5709 | 7.4103 | 38.6661 | 0.2062 |
| | SwinIR | 25.4564 | 0.7477 | 0.3818 | 0.2283 | 67.1467 | 6.9218 | 38.5181 | 0.2637 |
| | MambaIRv2 | 25.6781 | 0.6976 | 0.4686 | 0.2399 | 64.8715 | 7.4959 | 36.2515 | 0.2100 |
| | ISP+StableSR-s200 | 23.3339 | 0.6600 | 0.3505 | 0.1949 | 60.9322 | 3.9343 | 64.1478 | 0.6393 |
| | ISP+DiffBIR-s50 | 25.3643 | 0.6761 | 0.4086 | 0.2478 | 56.7401 | 5.6140 | 49.4878 | 0.5581 |
| | ISP+PASD-s20 | 24.8545 | 0.6886 | 0.3055 | 0.1720 | 40.8756 | 4.1290 | 63.5759 | 0.6223 |
| | ISP+SeeSR-s50 | 24.8332 | 0.6957 | 0.2872 | 0.1807 | 36.0702 | 4.2017 | 66.3191 | 0.6977 |
| | ISP+SUPIR-s50 | 23.9782 | 0.6505 | 0.3412 | 0.1937 | 51.7890 | 4.9086 | 59.3107 | 0.4814 |
| | ISP+OSEDiff-s1 | 23.8067 | 0.6872 | 0.2988 | 0.1768 | 52.0761 | 4.2011 | 65.5805 | 0.6793 |
| | Ours-s1 | 25.1264 | 0.7092 | 0.2546 | 0.1589 | 36.8671 | 4.1286 | 65.8881 | 0.6723 |

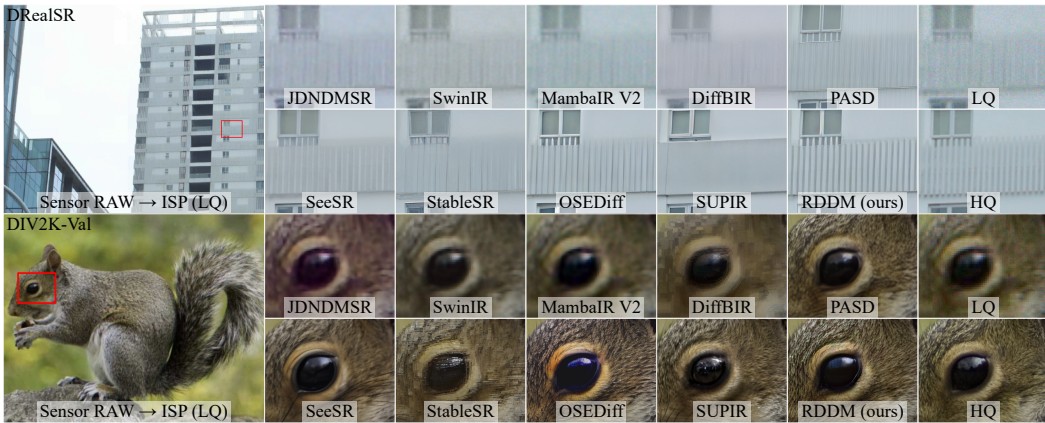

Figure 6: Qualitative comparison between RDDM and RAW domain one-stage method and two-stage ISP→IR methods on DRealSR dataset and DIV2K-val dataset.

## 4 EXPERIMENTS

### 4.1 EXPERIMENTAL SETTINGS

**Training and Testing Datasets.** We train RDDM using the LSDIR (Li et al., 2023) dataset and the first 10K face images from FFHQ (Karras et al., 2019). We use the degradation pipeline discussed in the Data Synthesis section to synthesize LQ and HQ pairs in the RAW domain. For testing, we use in-the-wild DND (Plotz & Roth, 2017) dataset, consists of 50 real noisy images captured by different

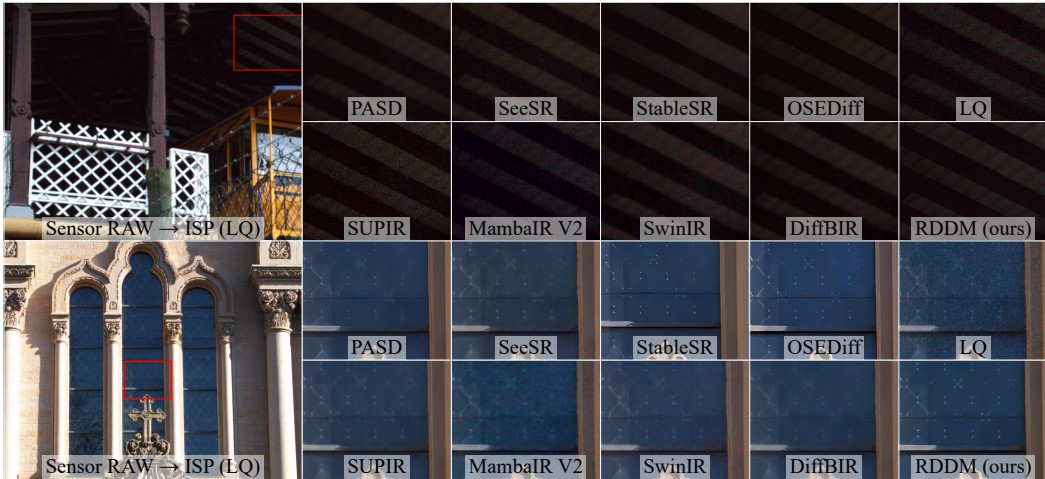

Figure 7: Qualitative comparison between RDDM and RAW domain one-stage method and two-stage ISP→IR methods on in-the-wild DND dataset.

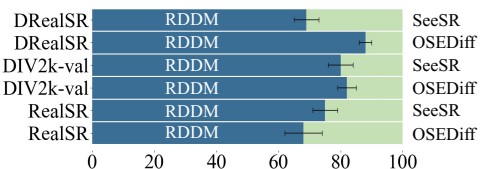

Figure 8: The user preference win rates of RDDM, compared to OSEDiff and SeeSR based on RealSR, DRealSR, and DIV2K-val. We provide the 95% confidence interval of the win rate based on five independent annotation rounds.

Table 2: Comparisons of Params and FLOPs between RDDM and its competing methods on input resolution of $512 \times 512$.

| Method | Params(M) | FLOPs(G) |
|---|---|---|
| JDnDmSR | 78.2 | 54 |
| SwinIR | 11.6 | 760 |
| MambaIRv2 | 31.4 | 8260 |
| ISP+StableSR-s200 | 1413 | 830 |
| ISP+DiffBIR-s50 | 1673 | 1670 |
| ISP+PASD-s20 | 1432 | 1590 |
| ISP+SeeSR-s50 | 1622 | 1230 |
| ISP+SUPIR-s50 | 4805 | 4100 |
| ISP+OSEDiff-s1 | 1298 | 250 |
| Ours-s1 | 1294 | 250 |

consumer cameras, including Sony A7R, Olympus OMD E-M10, Sony RX100 IV and Nexus 6P, and synthetic dataset construct by degrading the HR images from the DIV2K-Val, consisting of 100 images, RealSR containing 100 images, and DRealSR containing 93 images, using our proposed data synthesis method.

**Compared Methods.** We compare RDDM against best best-performing traditional one-stage method and two-stage ISP→IR models methods, as shown in Table 1. For one-stage methods, we re-train JDnDmSR (Xing & Egiazarian), SwinIR (Liang et al., 2021) and MambaIRv2 (Guo et al., 2025) on the same RAW domain training dataset as our baseline. For two-stage ISP→IR methods, we choose PIPNet (A Sharif et al., 2021) as the DN and DM module for ISP and diffusion-based IR methods as our Real-IR baselines, including StableSR (Wang et al., 2024), DiffBIR (Lin et al., 2024), PASD (Yang et al., 2024), SeeSR (Wu et al., 2024b), SUPIR (Yu et al., 2024) and OSEDiff (Wu et al., 2024a). Comparison results regarding GAN-based Real-IR methods, including BSRGAN Zhang et al. (2021), Real-ESRGAN Wang et al. (2021), LDL Liang et al. (2022a) and FeMaSR Chen et al. (2022), are provided in Appendix A.2.

**Evaluation Metrics.** For a thorough assessment of the different methods, we utilize a variety of full-reference and non-reference evaluation metrics to test each method's image fidelity and generation quality. PSNR and SSIM (Wang et al., 2004) (calculated on 3 channels) measure image fidelity, whereas LPIPS (Zhang et al., 2018a) and DISTS (Ding et al., 2020) measure perceptual qualities based on reference images. FID (Heusel et al., 2017) assesses the distributional distance between the GT and the restored images. NIQE (Mittal et al., 2012), MUSIQ (Ke et al., 2021), and CLIPIQA (Wang et al., 2023) are non-reference image generation quality measurements.

**Implementation Details.** We train RDDM with the AdamW optimizer at a learning rate of $5e-5$. The entire training process spans 150000 steps with a batch size of 16. The rank of LoRA in the

RVAE Encoder and the diffusion network is set to 4. We employ DAPE as the sRGB domain text prompts extractor.

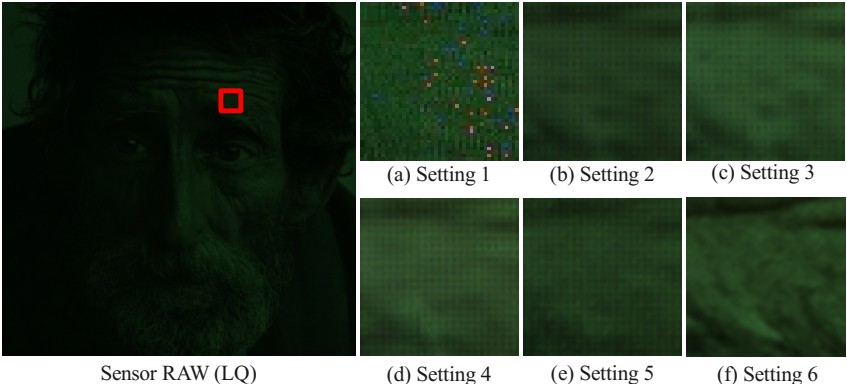

Figure 9: Qualitative comparison of different VAE settings on the RealSR benchmark. Setting 6 (ours) achieves the optimal performance.

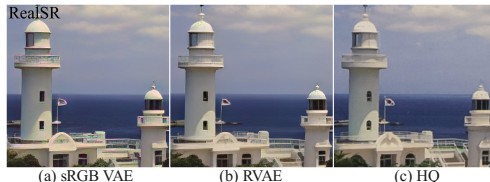

Figure 10: Qualitative comparison of sRGB VAE and RVAE on the RealSR benchmark.

Table 3: Quantitative reconstruction performance of different VAE settings on RealSR.

| Setting | PSNR↑ | SSIM↑ | LPIPS↓ | FID↓ |
|---------|-------|-------|--------|------|
| 1 | 25.2625 | 0.8017 | 0.1719 | 41.8118 |
| 2 | 27.1487 | 0.7035 | 0.3068 | 48.4675 |
| 3 | 27.6892 | 0.7176 | 0.2787 | 38.2007 |
| 4 | 26.3147 | 0.6703 | 0.3967 | 53.4004 |
| 5 | 25.4509 | 0.6277 | 0.4533 | 64.2444 |
| 6 | 32.5424 | 0.9082 | 0.0533 | 11.6156 |

Table 4: Quantitative performance of RDDM with different VAE settings on the RealSR dataset.

| Setting | PSNR↑ | SSIM↑ | LPIPS↓ | DISTS↓ | FID↓ | NIQE↓ | MUSIQ↑ | CLIPIQA↑ |
|---------|-------|-------|--------|--------|------|-------|--------|----------|
| 1 | 20.5609 | 0.5797 | 0.3930 | 0.2071 | 73.5659 | 4.0762 | 62.7875 | 0.6431 |
| 2 | 22.3752 | 0.6484 | 0.4785 | 0.3307 | 138.43 | 6.5993 | 47.2304 | 0.4233 |
| 3 | 21.0398 | 0.6115 | 0.5710 | 0.4235 | 143.3710 | 10.5422 | 37.3569 | 0.3745 |
| 4 | 21.3850 | 0.6080 | 0.4250 | 0.2901 | 108.3618 | 7.5271 | 52.8636 | 0.2931 |
| 5 | 21.3886 | 0.5732 | 0.4911 | 0.3166 | 135.6228 | 7.5537 | 38.7119 | 0.1483 |
| 6 | 25.1264 | 0.7092 | 0.2546 | 0.1589 | 36.8671 | 4.1286 | 65.8881 | 0.6723 |

Table 5: Comparison of different domain losses on the DIV2K benchmark.

| RAW Loss | sRGB Loss | PSNR↑ | SSIM↑ | LPIPS↓ | DISTS↓ | FID↓ | NIQE↓ | MUSIQ↑ | CLIPIQA↑ |
|----------|-----------|-------|-------|--------|--------|------|-------|--------|----------|
| ✓ | ✗ | 22.9510 | 0.6096 | 0.2794 | 0.1337 | 36.3172 | 3.4637 | 66.1391 | 0.7298 |
| ✗ | ✓ | 23.3129 | 0.6237 | 0.2635 | 0.1216 | 29.5131 | 3.3352 | 64.7748 | 0.6637 |
| ✓ | ✓ | 23.7416 | 0.6296 | 0.2540 | 0.1197 | 23.8028 | 3.3627 | 65.4202 | 0.6737 |

## 4.2 COMPARISONS WITH STATE-OF-THE-ARTS

**Quantitative Comparisons.** Table 1 presents the quantitative comparisons on three test datasets. RDDM ranks in the top 3 for all metrics, including PSNR, SSIM, LPIPS, DISTS, NIQE, MUSIQ, CLIPIQA, and FID, across DIV2K-Val, DRealSR, and RealSR, except for PSNR on RealSR. JD-nDmSR and SwinIR achieve slightly higher PSNR and SSIM on RealSR but significantly underperform on other metrics, particularly NIQE, MUSIQ, CLIPIQA, and FID, indicating weaker generative capabilities. RDDM matches diffusion-based methods in generative performance while outperforming them in image fidelity metrics like PSNR and SSIM. Additionally, RDDM has the lowest Params

and FLOPs among diffusion-based models, as shown in Table 2. More quantitative comparison results with GAN-based method are in the Appendix A.2.

**Qualitative Comparison.** Fig. 6 presents visual comparisons of RDDM on RealSR, DIV2K-val, and DRealSR versus RAW domain one-stage and two-stage methods. In the first example, JDnDmSR, SwinIR, DiffBIR and SUPIR generate blurry wall textures with JDnDmSR showing significant color deviation. PASD, SeeSR, StableSR, and OSEDiff produce more textures but lack fine details. RDDM produces realistic, high-clarity wall textures. The second example reinforces this conclusion, highlighting that RDDM utilizes detailed signal in sensor RAW resulting in mitigating artifacts in diffusion-based methods. More visualization comparison results are in the Appendix A.3. To further investigate the user preferences about these results, we conduct a user study on RealSR, DRealSR, and DIV2K-val test datasets, with 5 participants involved. For each set of comparison images, users select their preferred result. As shown in Fig. 8, the results demonstrate that our method significantly outperforms state-of-the-art methods in terms of perceptual quality. Additionally, experiments on in-the-wild camera dataset validate that our method outperforms other methods in terms of restoration capability under high-noise conditions, as shown in Fig. 7.

Table 6: Comparison of different text prompt extractors on the RealSR benchmark.

| | PSNR↑ | SSIM↑ | LPIPS↓ | DISTS↓ | FID↓ | NIQE↓ | MUSIQ↑ | CLIPIQA↑ |
|---|---|---|---|---|---|---|---|---|
| sRGBPE | 24.7811 | 0.7201 | 0.2495 | 0.1601 | 38.1092 | 4.2538 | 65.4506 | 0.6579 |
| ISPPE | 24.7816 | 0.7201 | 0.2495 | 0.1600 | 38.0969 | 4.2539 | 65.4526 | 0.6579 |
| RPE | 25.1264 | 0.7092 | 0.2546 | 0.1589 | 36.8671 | 4.1286 | 65.8881 | 0.6723 |

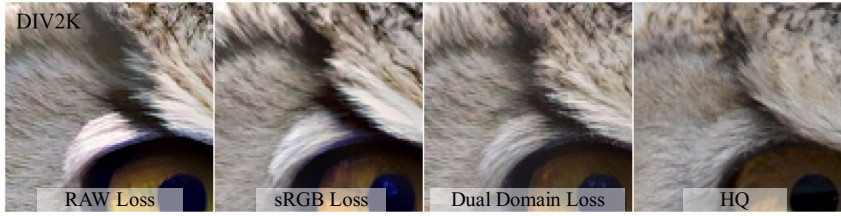

Figure 11: Qualitative comparison of RAW, sRGB and dual domain loss on the DIV2K-Val.

## 4.3 ABLATION STUDY

**The Importance of RVAE.** We evaluate six strategies for adapting VAEs to RAW domain reconstruction: (1) using a pre-trained sRGB VAE fails to reconstruct RAW data, producing poor linear domain images, as depicted in Fig. 9(a). (2) end-to-end VAE training with RAW-linear pairs results in mosaic textures, as illustrated in Fig. 9(b). (3) pre-training the encoder and decoder on linear images, then fine-tuning the encoder with LoRA on RAW-linear pairs, still produces mosaic textures, as displayed in Fig. 9(c). (4) replacing encoder LoRA with decoder LoRA in (3) retains mosaic textures, as shown in 9(d). (5) adding LoRA to both encoder and decoder in (3) still retains mosaic patterns, as presented in Fig. 9(e). (6) freezing the pre-trained decoder and jointly training the encoder and diffusion network with LoRA successfully recovers fine details and reduces color deviation, as demonstrated in Fig. 9(f). Table 3 shows that our method achieves the best reconstruction metrics, while Fig. 10 confirms that RVAE effectively reduces color deviation. Additionally, Table 4 demonstrate the end-to-end effect of different settings of RVAE within the RDDM framework and the result reinforces the optimality of our method in Setting 6. An ablation study on the CMB LoRA module reveals that a single LoRA shared across multiple Bayer patterns results in a slight performance degradation, as shown in Table 7.

**The Effectiveness of Dual Domain Loss.** We evaluate the impact of different domain losses in Table 5. Training RDDM solely in the RAW domain results in great image generation metrics but poor fidelity and perceptual quality, with localized color deviations. Conversely, training exclusively in the sRGB domain yields sub-optimal performance across all metrics but reduces color deviations. Training with supervision in both RAW and sRGB domains significantly enhances fidelity and per-

Table 7: Comparison of CMB LoRA and All-in-One LoRA on RealSR benchmark.

| | PSNR↑ | SSIM↑ | LPIPS↓ | DISTS↓ | FID↓ | NIQE↓ | MUSIQ↑ | CLIPIQA↑ |
|---|---|---|---|---|---|---|---|---|
| All-in-One LoRA | 24.5355 | 0.7003 | 0.2550 | 0.1555 | 40.6767 | 4.1165 | 65.0143 | 0.6660 |
| CMB-LORA | 25.1264 | 0.7092 | 0.2546 | 0.1589 | 36.8671 | 4.1286 | 65.8881 | 0.6723 |

ceptual quality, though some image generation metrics like MUSIQ and CLIPIQA slightly decline, while visual quality markedly improves. Fig. 11 demonstrates the visual performance.

**The Comparison of Text Prompt Extractors.** Table 6 compares three RAW prompt-extraction strategies: (1) direct sRGB-extractor use; (2) ISP-based conversion; (3) the proposed RPE with DDNet denoising-demosaicing and PTP. RPE achieves the best PSNR, DISTS, FID, NIQE, MUSIQ, and CLIPIQA by providing accurate prompts that effectively activate diffusion priors.

## 5 CONCLUSION

We propose RDDM starting from sensor RAW, a novel paradigm for Real-IR, empowered by RVAE, CMB LoRA module, and RAW data synthesis IR dataset. RDDM mitigates sub-optimal IR results in lossy sRGB domains by incorporating detail-rich information on RAW images. Although generating more realistic details, its fidelity is presently limited by the 8× RVAE down-sampling inherited from SD 2.1; a mosaic-aware down-sampling strategy will be pursued in future work.

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

## ETHICS STATEMENT

**Commitment to Ethical Standards.** This work was conducted in full compliance with the ICLR Code of Ethics. We have reviewed and adhered to the principles of responsible stewardship, scientific excellence, harm avoidance, fairness, respect for privacy, and confidentiality.

**Societal Impact and Human Well-being.** Our research focuses on improving real-world image quality through RAW-domain restoration. Potential benefits include enhanced medical imaging, safer autonomous-vehicle perception, and more accessible photography tools. We have explicitly considered possible misuses (e.g., creation of misleading imagery) and designed the model to preserve authenticity metadata, thus minimizing negative societal impact.

**Scientific Integrity.** All experimental results were obtained with reproducible pipelines. Datasets are publicly available or obtained under permissive licenses; we provide full configuration files, code, and pre-trained weights in the supplementary material. No data were fabricated or selectively reported.

**Fairness and Non-discrimination.** Training data span multiple camera brands, geographic regions, and demographic groups. We report performance across a mixture of noise levels to ensure the model does not disproportionately favor any single population or noise level.

**Privacy & Confidentiality.** No personally identifiable information (PII) was collected or processed. All human subjects data were either synthetic or already anonymized before use. Raw sensor data were stripped of EXIF fields that could reveal location or identity.

**Conflict of Interest.** The authors declare no financial or personal relationships that could influence the presented research.

**Compliance & Reporting.** Any future concerns regarding ethical compliance can be directed to the corresponding author at the institution listed on the title page.

## REPRODUCIBILITY STATEMENT

The code and complete reproduction protocols will be released upon publication.

## A  APPENDIX

In the appendix, we provide the following materials:

- Additional demonstrations regarding ISP and inverse ISP (referring to Section 3.1 in the main paper).
- Comparisons against GAN-based methods (referring to Section 4.1 in the main paper).
- More qualitative comparisons (referring to 4.2 in the main paper).

### A.1  APPENDIX: ISP AND INVERSE ISP

**ISP and Inverse ISP.** The RAW sensor data obtained by a camera is fundamentally different from the sRGB images which closely resemble human visual perception. Indeed, it is necessary to process RAW images through an ISP to obtain the final sRGB images. Fig. 5 illustrates the processing pipelines of ISP and InverseISP. We modify the InverseISP of Brooks et al. (2019) and propose a differentiable ISP approach that can map the model's output to the sRGB domain for optimization. During the ISP process, denoising and demosaicing are ill-posed problems, and we optimize these two tasks jointly with Real-IR task. Above all, in the ISP process, the Automatic White Balance (AWB) algorithm multiplies the red and blue channels by gains to produce an image that appears to be lit under "neutral" illumination, which can be formulated as Equation 8:

$$x_L^{awb} = X_L^{nm} \odot M_{gain}, \quad x_L^{nm} = X_L^{awb} \odot \frac{1}{M_{gain}} \tag{8}$$

where $x_L^{awb}$, $X_L^{nm}$ and $M_{gain}$ are the linear-domain image processed by AWB, the linear image processed by denoising and demosaicing, and the pixel-wise channel gain, respectively.

Secondly, the color correction algorithm converts its own "camera space" sRGB color measurements to sRGB values by a color correction matrix as shown in Equation 9:

$$x_L^{ccm} = x_L^{awb} \times M_c, \quad x_L^{awb} = x_L^{ccm} \times M_c^{-1} \tag{9}$$

where $x_L^{ccm}$, $M_c$ are the linear-domain image processed by CCM algorithm and color correction matrix, respectively.

Thirdly, since humans are more sensitive to gradations in the dark areas of images, gamma compression is typically used to allocate more bits of dynamic range to low-intensity pixels as stated in Equation 10:

$$X_L^{gamma} = max(x_L^{ccm}, \epsilon)^{1/2.2}, \quad X_L^{ccm} = max(x_L^{gamma}, \epsilon)^{2.2} \tag{10}$$

Note that we set $\epsilon = 10^{-8}$ to prevent numerical instability during training.

InverseISP is the inverse process of ISP, where the mosaicing algorithm acquires the RAW image $X_L^{RAW} \in \mathcal{R}^{H \times W \times 1}$ with a CFA by extracting the corresponding pixel values from the three channels. The noise in RAW images mainly comes from two sources: photon arrival statistics (shot noise) and imprecision in the readout circuitry (read noise). We can approximate these two types of noise as a single heteroscedastic Gaussian distribution defined in Equation 11:

$$y \sim \mathcal{N}(\mu = X_L^{mosaic}, \sigma^2 = \lambda_{shot} X_L^{mosaic} + \lambda_{read}) \tag{11}$$

The linear-domain image is obtained by Equation 12:

$$X_L^{lin} = X_L^{mosaic} + y \tag{12}$$

where $X_L^{mosaic}$ is the RAW image processed by mosaic algorithm and $y$ is the noise intensity added onto $X_L^{mosaic}$ to obtain the Sensor RAW $X_L^{RAW}$. $\lambda_{shot}$ and $\lambda_{read}$ are the function of ISO light sensitivity level.

## A.2 APPENDIX: QUANTITATIVE COMPARISONS WITH GAN-BASED METHODS

We compare RDDM against four two-stage ISP→GAN-based-IR methods. For the first ISP stage, we again use PIPNet (A Sharif et al., 2021) as the DN and DM module; for the second IR stage, we use BSRGAN Zhang et al. (2021), Real-ESRGAN Wang et al. (2021), LDL Liang et al. (2022a) and FeMaSR Chen et al. (2022) as our baselines. The results are shown in Table 8. Admittedly, two-stage methods involving GAN-based IR models demonstrate better fidelity results as expected, i.e. higher PSNR and SSIM metrics. However, their image generation capability is far behind RDDM. Visualization comparisons are shown in Fig. 12.

Table 8: Quantitative comparison with different methods on both synthetic benchmark. ↓ presents the smaller the better, ↑ presents the bigger the better. Our method RDDM exceeds its competing models in terms of image fidelity, whereas maintaining a satisfactory image generation ability approximately equivalent to the other baselines.

| Dataset | Method | PSNR↑ | SSIM↑ | LPIPS↓ | DISTS↓ | FID↓ | NIQE↓ | MUSIQ ↑ | CLIPIQA ↑ |
|---------|--------|-------|-------|--------|--------|------|-------|---------|-----------|
| DIV2K-Val | ISP+BSRGAN | **24.4124** | **0.6500** | 0.4174 | 0.2173 | 33.8937 | 5.1222 | 44.2923 | 0.4580 |
| | ISP+Real-ESRGAN | 24.2373 | 0.6426 | 0.3888 | 0.2007 | 35.3941 | 4.8658 | 48.6062 | 0.5157 |
| | ISP+LDL | 24.2922 | 0.6243 | 0.4684 | 0.2540 | 34.9855 | 5.5153 | 29.8086 | 0.3714 |
| | ISP+FeMaSR | 23.5889 | 0.6066 | 0.4347 | 0.2273 | 34.9390 | 5.7403 | 43.3786 | 0.5187 |
| | Ours | 23.7416 | 0.6296 | **0.2540** | **0.1197** | **23.8028** | **3.3627** | **65.4202** | **0.6737** |
| DRealSR | ISP+BSRGAN | **30.1134** | **0.8282** | 0.3196 | 0.1982 | 20.5962 | 5.6748 | 42.4667 | 0.5105 |
| | ISP+Real-ESRGAN | 29.1309 | 0.8055 | 0.3361 | 0.2042 | 23.5094 | 5.5440 | 46.0976 | 0.5548 |
| | ISP+LDL | 29.9312 | 0.7967 | 0.3640 | 0.2173 | 20.9155 | 5.9973 | 29.3485 | 0.4047 |
| | ISP+FeMaSR | 28.1176 | 0.7460 | 0.4007 | 0.2356 | 24.3254 | 6.3530 | 41.3755 | 0.5844 |
| | Ours | 28.3495 | 0.7892 | **0.2719** | **0.1649** | **17.4825** | **4.6852** | **57.0696** | **0.7035** |
| RealSR | ISP+BSRGAN | **27.0999** | **0.7648** | 0.3190 | 0.1984 | 50.7266 | 5.0958 | 50.2340 | 0.5052 |
| | ISP+Real-ESRGAN | 26.3844 | 0.7474 | 0.3332 | 0.1995 | 49.3939 | 4.5754 | 55.7670 | 0.5498 |
| | ISP+LDL | 26.5426 | 0.7186 | 0.3871 | 0.2141 | 53.1533 | 5.2573 | 33.9660 | 0.3656 |
| | ISP+FeMaSR | 25.5820 | 0.6898 | 0.3891 | 0.2278 | 53.7582 | 5.8086 | 51.1097 | 0.5917 |
| | Ours | 25.1264 | 0.7092 | **0.2546** | **0.1589** | **36.8671** | **4.1286** | **65.8881** | **0.6723** |

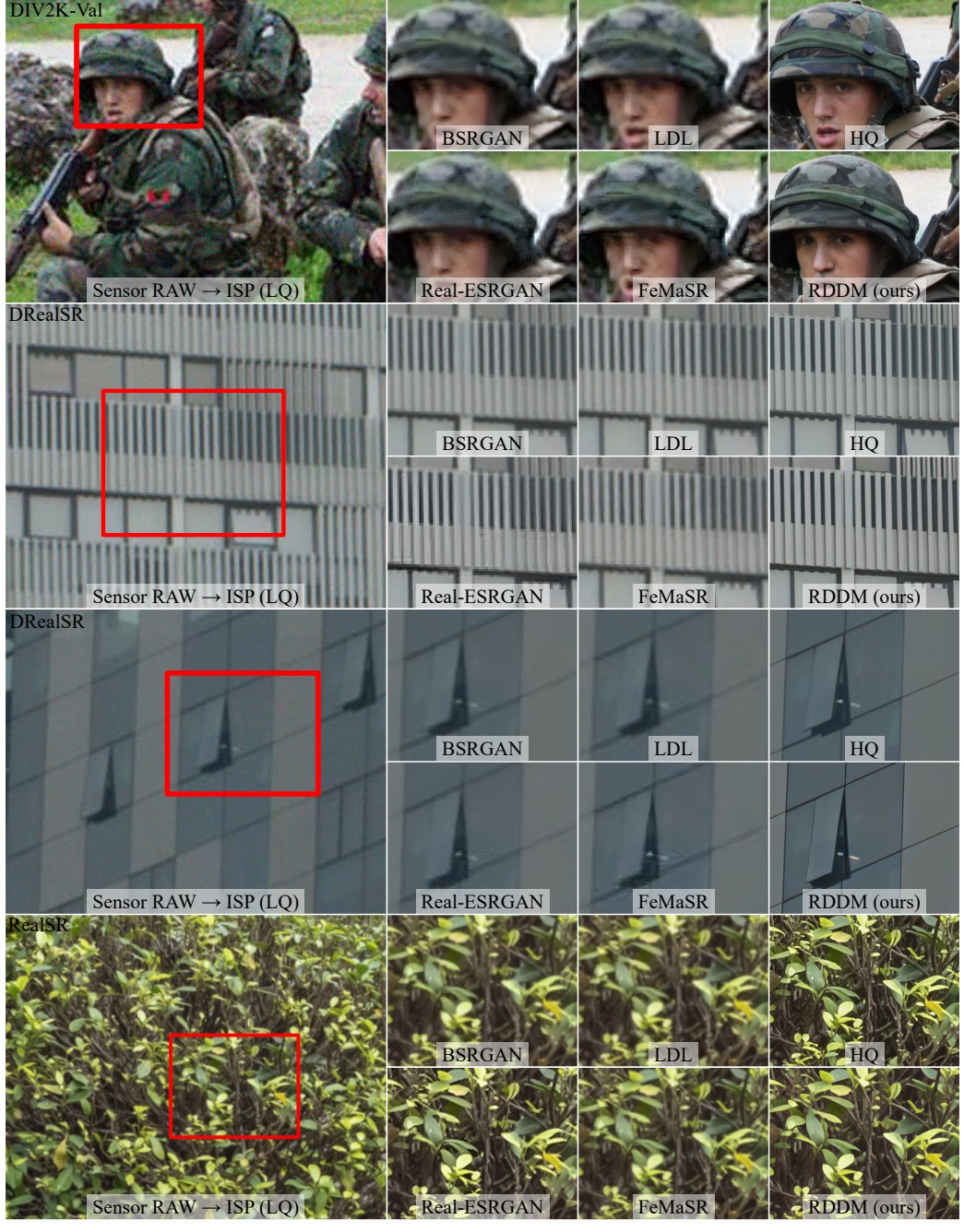

Figure 12: Qualitative comparison between RDDM and GAN-based methods on DIV2K-val, DRealSR and RealSR dataset.

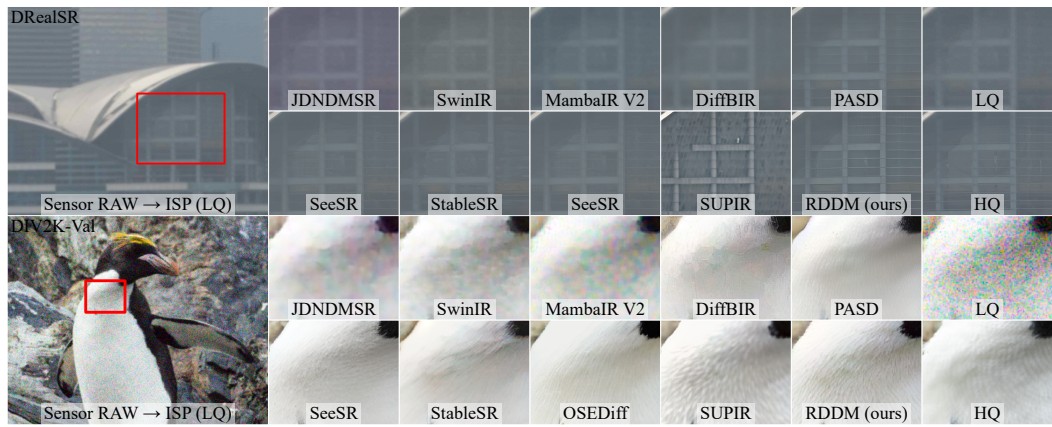

Figure 13: Qualitative comparison between RDDM and RAW domain one-stage method and two-stage ISP→IR methods on DRealSR dataset and DIV2K-val dataset.

## A.4 APPENDIX: EXAMPLES OF SYNTHETIC RAW DATA

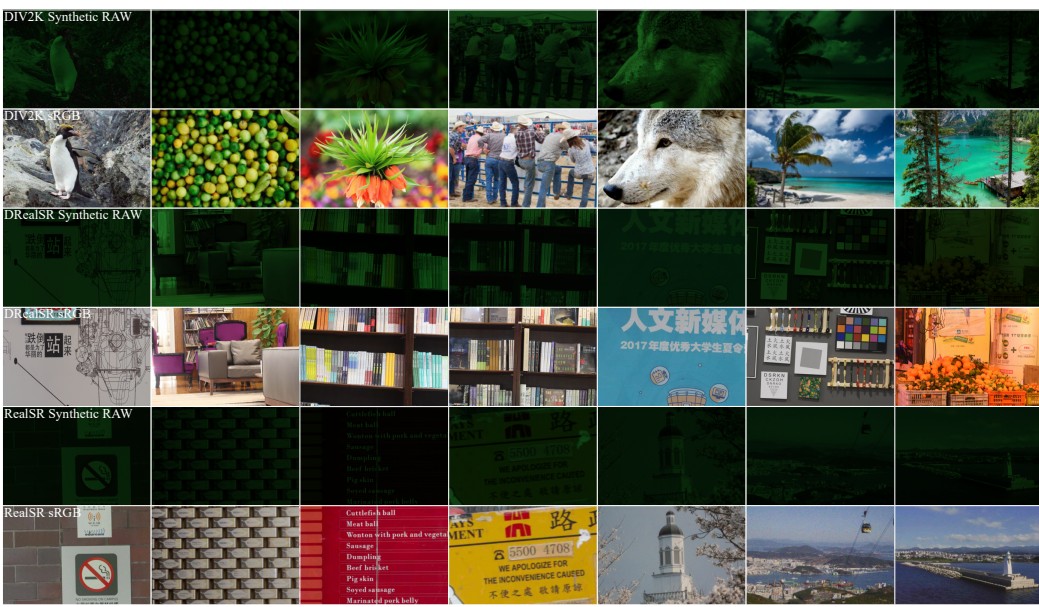

Figure 14: Examples of synthetic RAW data produced by our data synthesis pipeline. From top to bottom, each row represents the RAW data from DIV2K, the corresponding sRGB data from DIV2K, the RAW data from DRealSR, the corresponding sRGB data from DRealSR, the RAW data from RealSR and the corresponding sRGB data from RealSR.

