# OpenReview forum: "RDDM: Practicing RAW Domain Diffusion Model for Real-world Image Restoration"
_ICLR.cc/2026/Conference — ICLR 2026 Conference Withdrawn Submission_

### Official Review · Reviewer_NuQu · 2025-10-31

**Soundness:** 3
**Presentation:** 3
**Contribution:** 3
**Rating:** 4
**Confidence:** 5

**Summary:**

The paper introduces RDDM, a diffusion model that restores images directly from sensor RAW data, avoiding the lossy ISP→sRGB process. It employs a RAW Variational Autoencoder (RVAE) for RAW–linear mapping, a Configurable Multi-Bayer LoRA for different sensor patterns, and a RAW data synthesis pipeline for training. With a dual-domain loss combining RAW and sRGB supervision, RDDM achieves higher fidelity and perceptual quality than state-of-the-art methods, establishing a new paradigm for RAW-domain image restoration.

**Strengths:**

1. The paper is highly original in proposing the first RAW-domain diffusion model (RDDM) for real-world image restoration. Unlike prior works limited to the sRGB domain, it innovatively leverages unprocessed sensor RAW data, capturing richer signal information and eliminating dependence on handcrafted ISP pipelines. The integration of diffusion priors into the RAW domain represents a clear conceptual advance.
2. The technical design is solid and well-justified. The introduction of the RAW Variational Autoencoder (RVAE), Configurable Multi-Bayer LoRA, and RAW data synthesis pipeline effectively addresses domain gap and data scarcity issues. Comprehensive experiments on multiple benchmarks confirm the method’s superiority in both fidelity and perceptual quality, supported by strong quantitative and visual evidence.
Significance:
3. This work establishes a new paradigm for real-world image restoration by moving from the conventional sRGB domain to the sensor RAW domain. Its framework and data synthesis approach are broadly applicable to other low-level vision tasks and could significantly influence future research on RAW-domain generative modeling and camera imaging pipelines.

**Weaknesses:**

1. While RDDM shows strong results on several benchmarks, the experiments mainly focus on synthetic or semi-synthetic datasets derived from sRGB sources. The paper would be stronger with more evaluation on diverse real RAW datasets captured by different sensors and under varied lighting conditions to verify cross-camera robustness.
2. The proposed RAW data synthesis pipeline is creative but may not fully capture real-world sensor noise characteristics or color responses. The paper lacks quantitative analysis on how well the synthetic RAW data matches real RAW distributions, which could affect the model’s practical reliability.
3. The ablation study, while detailed, could include comparisons with simpler RAW-based baselines (e.g., RAW-ISP pipelines using modern neural ISPs) to better quantify the advantage of diffusion-based modeling in the RAW domain.

**Questions:**

1. Could the authors provide quantitative or visual comparisons between results trained on real RAW data (if available) and synthetic RAW data generated by their pipeline? This would help assess how well the synthetic data distribution aligns with real-world sensor characteristics.
2. How does RDDM perform when applied to RAW data from cameras with unseen Bayer patterns or sensor characteristics (e.g., mobile vs. DSLR sensors)? Have the authors tested the adaptability of the CMB-LoRA module in truly cross-device scenarios?
3. The dual-domain loss design is interesting. Could the authors clarify the relative contributions of RAW-domain and sRGB-domain supervision? For instance, what happens when each component’s weight (( \lambda_1, \lambda_2 )) is varied, or when the loss is applied asymmetrically?
4. Failure Cases and Visual Artifacts:
Could the authors show examples where RDDM fails or produces artifacts, and analyze the causes (e.g., misdemosaicing, over-smoothing, or hallucinated textures)? This would give a clearer picture of its limitations and possible improvement directions.

---

> ### Author Response · Authors · 2025-12-01
>
> Thank you for your thorough review and insightful suggestions. We would like to respond to the following concerns and questions:
>
> **Answering Weakness 1**. We already evaluated RDDM on real-world sensor RAW dataset DND (Darmstadt Noise Dataset) and the qualitative results are shown in Fig.7. For more real RAW datasets captured by different sensors and under varied lighting conditions, we will supplement more evaluations on SIDD [1] into our rebuttal revision version shortly.
>
> **Answering Weakness 2 & Question 1**. Our synthetic data for training contain the real-world distribution for sensor RAW. The read and shot noise added to our synthetic data is sampled from real-world sensor data. This implies that the distribution of our synthetic RAW data is similar to the real-world sensor RAW distribution, referring to Fig. 5 in UPI [2]. In addition, as explained earlier we evaluated RDDM on real-world sensor RAW data and the qualitative result demonstrate RDDM's effective generalization ability. Finally, in the real-world applications of RDDM, the synthetic degradation dataset can be customized by calibrating it for the specific sensor in use.
>
> **Answering Weakness 3**. Thank you for pointing this out. We will supplement the quantitative and qualitative results of some RAW-ISP methods in a later version.
>
> **Answering Question 2**. We will supplement quantitative and qualitative results of RDDM on SIDD [1] containing mobile-phone captured sensor RAW data, e.g. Google Pixel, iPhone 7 and Samsung Galaxy S6 Edge. Theoretically, CMB-LoRA is not designed to adapt to unseen bayer-patterns and has to be trained on the specific pattern for inference. However, a unified LoRA trained on mixed bayer patterns can adapt to multiple patterns. Please check **A3** in our official comments to reviewer VdBX for more insights about a unifed LoRA. Nevertheless, we provide the qualitative results of inferencing bggr RAW benchmark with CMB-LoRA trained on rggb pattern.
>
> **Answering Question 3**. Thank you for this insightful question. We will try our best to conduct a more precise ablation for the weights of RAW-domain and sRGB-domain. However, due to limited time and resources, the experiment might be available in a later version.
>
> **Answering Question 4**. The failure cases will be provided in the Appendix of our rebuttal revision version. Please check for more information and explanations.
>
> ## References
>
> [1] A. Abdelhamed, S. Lin, and M. S. Brown, *A high-quality denoising dataset for smartphone cameras*. In *IEEE/CVF Conference on Computer Vision and Pattern Recognition*, 2018, pp. 1692–1700.
>
> [2] Brooks, T., Mildenhall, B., Xue, T., Chen, J., Sharlet, D., & Barron, J. T., *Unprocessing images for learned raw denoising*. In *Proceedings of the IEEE/CVF Conference on Computer Vision and Pattern Recognition*, 2019, pp. 11036–11045.

---

### Official Review · Reviewer_BPDr · 2025-10-31

**Soundness:** 2
**Presentation:** 1
**Contribution:** 3
**Rating:** 2
**Confidence:** 3

**Summary:**

The paper proposes RDDM, a diffusion model that performs image restoration in the RAW domain rather than sRGB space. Directly adapting diffusion models to the RAW space presents out-of-distribution (OOD) issues. To overcome this issue, the authors propose a RAW domain VAE and a scalable data synthesis pipeline to create RAW images from existing sRGB data. The method achieves better perceptual quality and higher fidelity than state-of-the-art methods.

**Strengths:**

1. The proposed idea of using latent diffusion models for RAW domain restoration is interesting.
2. The qualitative results have more detail and higher fidelity than existing methods.

**Weaknesses:**

1. Hard to understand: The flow and explanation of the methodology section is very difficult to follow (see questions).
2. Unclear motivation: From my understanding, the paper proposes a method that restores a LQ RAW image ($X_L^\text{RAW}$) to its HQ counterpart in RGB space ($X_H^\text{rgb}$). Existing methods directly perform restoration on the LQ image in RGB space. The methodology in the paper depends on converting the LQ image from RGB space to the RAW space and then performing restoration (both in training and inference). However, isn’t this inverse mapping not perfect as RAW to RGB mapping is lossy? Then, the performance of the entire method depends on how well this inverse mapping is learnt right? There are no experiments validating the effectiveness of the inverse mapping, which makes it unclear as to why the proposed methodology is working.
3. No experiments validating the effectiveness of the data synthesis pipeline.

**Questions:**

1. What is the difference between linear domain and RAW domain? There is no mention of this in Sec. 3.1.
2. How are $\mathcal{F}_\text{DD}$ and $\mathcal{F}_\text{PTP}$ trained? Sec. 3.1 assumes the presence of data pairs $(X_L^\text{RAW}, X_H^\text{lin}$, but there is no such available data right (only HQ RAW and RGB images are available?)?
3. What is the dataset used to train the VAE?
4. Most degradations in LQ RGB images are caused by compression, blur, noise, etc. These degradations are prevalent in RGB post-processing techniques (for instance JPEG compression) and would be absent in RAW sensor data. So, is it fair to consider the RAW input image as an equivalent LQ in RGB space, which existing SOTA approaches work on?
5. How are $\mathcal{F}^{-1}_\text{DD}$ and $\mathcal{F}^{-1}_\text{PTP}$ trained?
6. Can the authors provide examples of synthesized data?

---

> ### Author Response · Authors · 2025-11-28
>
> Thank you for your reviews. We would like to respond to the following concerns and questions.
>
> **Answering Weakness 2&3**. Our motivation is to boost image restoration quality on edge-device imaging by unleashing the full potential of diffusion models in the RAW domain. In the real world scenario, sensor RAW image data contain more information than the sRGB image processed by ISP, elevating the upper limit of the diffusion-based IR models. Hence, delivering the first diffusion model in the RAW domain can pioneer the new trend in edge-device imaging, manifested by our other reviewer NuQu. However, large scale real RAW image restoration data is hardly publically available. Therefore, a RAW data synthesis pipeline is necessary and widely adopted, e.g. by [1], to simulate real RAW data and enable large scale training. The generated RAW data have similar distribution with the real data, endorsed by [1]. In addition, the fact that RDDM trained on data generated by our synthesis pipeline generalizes well to real RAW data further manifests the effectiveness of our pipeline. Finally, for RDDM and its competing two-stage ISP-->IR methods, we used the same RAW benchmark, i.e. the same input, to evaluate their performance. However, RDDM restores and generates in the RAW domain whereas the two-stage ISP-->IR baselines first convert RAW data to a lossy sRGB image and then restore the lossy sRGB image to obtain HQ. Therefore, RDDM restoring in the more informative RAW domain is superior than those restoring in the sRGB domain.
>
> **Answering Q1**. In short, RAW domain is a specific subset of linear domain. Linear domain images with mosaic patterns and sensor-specific original noises are RAW images. To be more specific, edge-devices capture sensor RAW data. After demosaicing module of ISP, a RAW domain image is transformed into a linear domain image. After post tone processing (e.g. luminance or color correction), a linear domain image is transformed into an sRGB domain image. Thank you for pointing out this question and we have supplemented this background knowledge in our related work.
>
> **Answering Q2&Q5**.  $\mathcal{F}\_{\text{PTP}}$ is a forward tone mapping pipeline and is train-less. $\mathcal{F}^{-1}\_{\text{PTP}}$ is the inverse $\mathcal{F}\_{\text{PTP}}$.  $\mathcal{F}\_{\text{DD}}$ is a light-weight CNN-based joint denoising and demosaicing net trained with RAW-linear image pairs degraded from LSDIR and ffhq through our proposed data synthesis pipeline, following the the training strategy of [2]. $\mathcal{F}^{-1}_{\text{DD}}$ is part of the data synthesis pipeline degrading linear images to RAW images by adding noises and mosaics, which is train-less.
>
> **Answering Q3**. RVAE is trained with linear domain image pairs degraded from DIV2K, Flickr2K, LSDIR, DIV8K and the first 10K face images from FFHQ. We have suppemented this information in "Training and Testing Datasets" in Sec. 4.1.
>
> **Answering Q4**. As mentioned in answers for **Weakness 2&3**, we would like to reiterate that the RAW-domain benchmarks we used for all the baseline methods, including two-stage ISP-->IR methods and our RDDM, are exactly the same. It is a fair comparison between the baselines and our methods. The only difference is that, for two-stage ISP-->IR methods, the RAW-domain benchmark has to go through ISP to become an sRGB image for the subsequent sRGB-domain RealIR methods (e.g. StableSR, DiffBIR, PASD, SeeSR, SUPIR and OSEDiff) to perform image generation etc., whereas RDDM directly performs restoration and image generation on RAW images. Hence, two-stage methods that require an intermediate lossy sRGB input is exactly their drawback, and is exactly our motivation to propose a diffusion model in the RAW domain, avoiding any intermediate lossy sRGB inputs.
>
> **Answering Q6**. Some synthetic RAW data are supplemented in the Appendix Section A.4 of our rebuttal revision version.
>
> ## References
> [1] Brooks, T., Mildenhall, B., Xue, T., Chen, J., Sharlet, D., & Barron, J. T. (2019). *Unprocessing images for learned raw denoising*. In *Proceedings of the IEEE/CVF Conference on Computer Vision and Pattern Recognition* (pp. 11036–11045).
>
> [2] Sharif, S. M. A., Naqvi, R. A., & Biswas, M. (2021). Beyond joint demosaicking and denoising: An image processing pipeline for a pixel-bin image sensor. In *Proceedings of the IEEE/CVF Conference on Computer Vision and Pattern Recognition* (pp. 233–242).

---

### Official Review · Reviewer_VdBX · 2025-11-01

**Soundness:** 2
**Presentation:** 2
**Contribution:** 2
**Rating:** 2
**Confidence:** 4

**Summary:**

This paper presents the RAW Domain Diffusion Model (RDDM), a novel approach for image restoration that operates directly on sensor RAW data, bypassing the conventional two-stage ISP-then-restoration pipeline.  The technical solutions (RVAE, CMB-LoRA) appear to directly address the core challenges of OOD and pattern diversity.

**Strengths:**

1. Sufficient presentation of objective metrics and subjective visualization results in the experiments:
2. Clear description of the methodology and contributions:

**Weaknesses:**

1. As shown in Table 1, many results in the comparison with SOTA methods are suboptimal. Does this imply that the current solution still has limitations, and that more relevant analysis and experiments should be provided?
2. It is important to note that a VAE's optimal reconstruction results are not indicative of superior generation capability, given the frequent trade-off between these two objectives. Therefore, in addition to the reconstruction performance presented in Table 3, an evaluation of the generation performance is necessary to conclusively determine a better VAE structure.
3. As shown in Figure 4, different input modes require LoRAs with different parameters, which reduces overall practicality. Could a unified LoRA be considered to achieve this functionality without switching?

**Questions:**

1. Some information across the tables appears inconsistent, such as the method names in Figure 1 and the configuration codes in Figure 9. More complete information should be provided in the captions to facilitate reading.
2. Several minor errors in the text need correction to improve readability, such as the garbled characters appearing in line 466.

---

> ### Author Response · Authors · 2025-11-28
>
> Thank you for your review. We have revised all the typos and unclear captions, which can be found in our rebuttal revision version. We would like to respond to the following reviews:
> > **Q1**. As shown in Table 1, many results in the comparison with SOTA methods are suboptimal. Does this imply that the current solution still has limitations, and that more relevant analysis and experiments should be provided?
>
> **A1:** In Table 1., RDDM demonstrates significantly better image fidelity quality whereas maintaining a comparable image generation quality when compared against diffusion-based two-stage methods. When compared against transformer-based one-stage methods, RDDM demonstrates comparable image fidelity quality whereas the baseline methods have almost no image generation quality. Qualitatively, RDDM demonstrates better perceptual quality than both one-stage and two-stage methods as shown in Fig. 6. Therefore, our paradigm of diffusion model in the RAW domain is experimented to be effective. However, part of the original information on RAW data is lost due to the downsampling strategy intrinsic to the RVAE, which indeed is the limitation of RDDM. A stronger RVAE can further boost the RDDM's performance.
>
> > **Q2**. It is important to note that a VAE's optimal reconstruction results are not indicative of superior generation capability, given the frequent trade-off between these two objectives. Therefore, in addition to the reconstruction performance presented in Table 3, an evaluation of the generation performance is necessary to conclusively determine a better VAE structure.
>
> **A2:** Thank you for this suggestion. We exprimented the end-to-end effect of different settings of RVAE in the RDDM framework and the quantitative results are pasted below. The same table is updated in our rebuttal revision version in Table 4. The results demonstrate that our method (setting 6) is superior to all other settings of RVAE. A better reconstruction ability of RVAE is key to the success of the image fidelity and generation quality of the subsequent joint training.
>
> | Setting | PSNR↑  | SSIM↑  | LPIPS↓ | DISTS↓ | FID↓    | NIQE↓  | MUSIQ↑ | CLIPIQA↑ |
> |---------|--------|--------|--------|--------|---------|--------|--------|----------|
> | 1       | 20.5609| 0.5797 | 0.3930 | 0.2071 | 73.5659 | 4.0762 | 62.7875| 0.6431   |
> | 2       | 22.3752| 0.6484 | 0.4785 | 0.3307 | 138.43  | 6.5993 | 47.2304| 0.4233   |
> | 3       | 21.0398| 0.6115 | 0.5710 | 0.4235 | 143.3710| 10.5422| 37.3569| 0.3745   |
> | 4       | 21.3850| 0.6080 | 0.4250 | 0.2901 | 108.3618| 7.5271 | 52.8636| 0.2931   |
> | 5       | 21.3886| 0.5732 | 0.4911 | 0.3166 | 135.6228| 7.5537 | 38.7119| 0.1483   |
> | 6       | 25.1264| 0.7092 | 0.2546 | 0.1589 | 36.8671 | 4.1286 | 65.8881| 0.6723   |
>
> *Table: Quantitative performance of RDDM with different VAE settings on the RealSR dataset.*
>
>
> > **Q3**. As shown in Figure 4, different input modes require LoRAs with different parameters, which reduces overall practicality. Could a unified LoRA be considered to achieve this functionality without switching?
>
> **A3:** We considered a unified LoRA when designing our paradigm and the unified LoRA generated decent perceptual images. However, its performance is sligntly less optimal than the current pattern-specific CMB-LoRA with same rank (rank = 4). Table below shows the optimality of CMB-LoRA. The same table is updated in Table 7 with explanations in our rebuttal revision version.
> |                  | PSNR↑  | SSIM↑  | LPIPS↓ | DISTS↓ | FID↓   | NIQE↓  | MUSIQ↑ | CLIPIQA↑ |
> |------------------|--------|--------|--------|--------|--------|--------|--------|----------|
> | All-in-One LoRA  | 24.5355| 0.7003 | 0.2550 | 0.1555 | 40.6767| 4.1165 | 65.0143| 0.6660   |
> | CMB-LORA         | 25.1264| 0.7092 | 0.2546 | 0.1589 | 36.8671| 4.1286 | 65.8881| 0.6723   |
>
> *Table: Comparison of CMB LoRA and All-in-One LoRA on RealSR benchmark.*

---

### Official Review · Reviewer_z4Nw · 2025-11-01

**Soundness:** 2
**Presentation:** 1
**Contribution:** 2
**Rating:** 2
**Confidence:** 4

**Summary:**

The paper “RDDM: Practicing RAW Domain Diffusion Model for Real-World Image Restoration” presents an end-to-end diffusion framework that directly restores high-quality RGB images from sensor RAW data, bypassing traditional ISP (image signal processing) pipelines. The method introduces several components: a RAW-domain VAE (RVAE) for encoding mosaicked signals, a configurable multi-Bayer (CMB) LoRA for different sensor patterns, a dual-domain (RAW + sRGB) loss, and a RAW data synthesis pipeline for training data creation. Experiments on multiple benchmarks (DIV2K, RealSR, DRealSR, DND) demonstrate competitive results, suggesting advantages in both fidelity and perceptual quality compared to sRGB-domain diffusion and ISP-based two-stage baselines.

**Strengths:**

+ The combination of RAW-domain VAE, LoRA-based Bayer adaptation, and dual-domain supervision reflects careful engineering to handle distribution gaps and data diversity in sensor formats.
+ The authors provide a broad set of quantitative, qualitative, and user studies, covering both synthetic and in-the-wild datasets, and analyze component effectiveness through ablations on losses, VAE variants, and prompt extractors.

**Weaknesses:**

- Overly complicated and fragmented pipeline. The proposed system consists of multiple coupled modules. The overall architecture is highly procedural and difficult to apply or reason about conceptually. Much of the technical novelty lies in stitching existing components together rather than introducing a coherent or elegant new formulation.

- Limited conceptual novelty. The work mainly transfers existing ideas (latent diffusion, LoRA, ISP simulation, prompt-based conditioning) into the RAW domain. While this is a meaningful application, it is not a fundamental algorithmic advance in diffusion modeling or image restoration. The contributions are more empirical and system-engineering in nature than theoretical or methodological.

- Ambiguity in causal effectiveness and dependence on pretrained modules. It remains unclear how much of the reported improvement stems from genuine RAW-domain learning versus strong pretrained priors. Moreover, the dual-domain supervision appears heuristic and lacks deeper justification.

- There are too many typos in the paper, which significantly lowers the presentation and clarity of this work. For example, Line-093:  “enahnced linear domain image” → enhanced linear domain image; Line-268: "to obtain degraed RAW images” → degraded RAW images; Line-466: "as shown in Table ??."; Line-473: “Fig. 11 demonstrate the visual performance.” → “Fig. 11 demonstrates the visual performance.”, Line-484: “Although generating more realistic details, Its fidelity …” → “Although generating more realistic details, its fidelity …”, etc.

**Questions:**

How robust is RDDM to variations in real sensor calibration, non-Bayer mosaics, or camera-specific noise models that differ from the synthesized RAW data?

---

> ### Author Response · Authors · 2025-11-28
>
> Thank you for your review. We have revised all the typos and updated in our rebuttal revision version. We would like to respond to the following concerns:
>
> > **Q1**. Overly complicated and fragmented pipeline. The proposed system consists of multiple coupled modules. The overall architecture is highly procedural and difficult to apply or reason about conceptually. Much of the technical novelty lies in stitching existing components together rather than introducing a coherent or elegant new formulation.
>
> > **Q2**. Limited conceptual novelty. The work mainly transfers existing ideas (latent diffusion, LoRA, ISP simulation, prompt-based conditioning) into the RAW domain. While this is a meaningful application, it is not a fundamental algorithmic advance in diffusion modeling or image restoration. The contributions are more empirical and system-engineering in nature than theoretical or methodological.
>
> **A1&2**: We would like to reiterate that diffusion model starting from RAW images is a paradigm innovation that can profoundly impact edge-device imaging scenario, e.g. mobile phones, where RAW images are captured. Our conceptual novelty and technical design is recognized by our reviewer NuQu in the listed strengths 1~3. More specifically, edge-device sensors capture RAW images, which are converted to sRGB images through ISP for human-eye visualization. RAW images contain more information than sRGB images because ISP is a lossy process. Thus, restoring and generating images in the RAW domain elevates the performance ceiling of the diffusion models, whereas all existing diffusion-based models are trained and developed in the sRGB domain. Through extensive experiments, we found that there exist multiple problems with a naive adaptation of diffusion-based models and their relevant components to the RAW domain: 1. poor image restoration quality; 2. color shift etc. After careful theoretical analysis, we found that existing VAE pretrained in the sRGB domain cannot reconstruct RAW images. Hence, problem 1 and 2 are caused by domain mismatch between RAW and sRGB domain. For example, RAW domain images (1-channel) contain mosaic bayer patterns with pixel values linearly proportional to the luminance level, completely different from sRGB domain images (3-channel) after luminance and color correction. Therefore, we propose an RVAE that input RAW images and output 3-channel linear images along with a corresponding training strategy to boost its representation ability. To solve all the challenges during the design of RDDM requires intensive engineering, although a diffusion process starting from RAW is undeniabally a novel paradigm innovation. Finally, we will reorganize our framework and pipelines to highlight the essence of our paradigm, novelty and contribution.
>
> > **Q3**. Ambiguity in causal effectiveness and dependence on pretrained modules. It remains unclear how much of the reported improvement stems from genuine RAW-domain learning versus strong pretrained priors. Moreover, the dual-domain supervision appears heuristic and lacks deeper justification.
>
> **A3**: Regarding point 3, Table 1. in our paper demonstrates the effectiveness of RAW-domain learning. Specifically, we compared our RDDM against two-stage ISP->IR methods on the same benchmarks and these methods, e.g. ISP->OSEDiff, ISP->SUPIR and ISP->SeeSR, contain pre-trained priors. Please note that for two-stage ISP->IR baselines, the generative models (e.g. StableSR, DiffBIR, PASD, SeeSR, SUPIR and OSEDiff) are learned in the sRGB domain whereas for RDDM, the generative model is learned in the RAW domain. Our Table 1. demonstrates that our method has superior image fidelity and comparable image generation quality when compared with two-stage ISP->IR methods, indicating the effectiveness of RAW-domain learning.
>
> > **Q4**. How robust is RDDM to variations in real sensor calibration, non-Bayer mosaics, or camera-specific noise models that differ from the synthesized RAW data?
>
> **A4**: We already provide RDDM's qualitative results on real RAW benchmark (DND) in Fig. 7. The results show that our model generalizes well to the real sensor calibration and camera-specific noise models, e.g. Sony A7R and Olympus OMD E-M10. In addition, the noise (read and shot noise) added in our synthesized data is sampled from real-world distribution so that our synthesized data distribution contain real-world distribution. Please note that all RAW images are Bayer-mosaicked and non-Bayer mosaics don't occur in RAW images.

---

### Note · Authors · 2026-01-11

**Comment:**

We would like to thank the reviewers for their constructive feedback. We have decided to withdraw the submission at this time to allow ourselves enough time to conduct a major revision and restructuring of the paper based on the suggestions received.

**Withdrawal Confirmation:**

I have read and agree with the venue's withdrawal policy on behalf of myself and my co-authors.